# Rapeseed Flower Counting Method Based on GhP2-YOLO and StrongSORT Algorithm

**DOI:** 10.3390/plants13172388

**Published:** 2024-08-27

**Authors:** Nan Wang, Haijuan Cao, Xia Huang, Mingquan Ding

**Affiliations:** The Key Laboratory for Quality Improvement of Agricultural Products of Zhejiang Province, College of Advanced Agricultural Sciences, Zhejiang A&F University, Hangzhou 311300, China; jerrywang1010@sina.com (N.W.); chj102920@163.com (H.C.); 17318820085@163.com (X.H.)

**Keywords:** deep learning, rapeseed flower, StrongSORT, target tracking counting

## Abstract

Accurately quantifying flora and their respective anatomical structures within natural ecosystems is paramount for both botanical breeders and agricultural cultivators. For breeders, precise plant enumeration during the flowering phase is instrumental in discriminating genotypes exhibiting heightened flowering frequencies, while for growers, such data inform potential crop rotation strategies. Moreover, the quantification of specific plant components, such as flowers, can offer prognostic insights into the potential yield variances among different genotypes, thereby facilitating informed decisions pertaining to production levels. The overarching aim of the present investigation is to explore the capabilities of a neural network termed GhP2-YOLO, predicated on advanced deep learning techniques and multi-target tracking algorithms, specifically tailored for the enumeration of rapeseed flower buds and blossoms from recorded video frames. Building upon the foundation of the renowned object detection model YOLO v8, this network integrates a specialized P2 detection head and the Ghost module to augment the model’s capacity for detecting diminutive targets with lower resolutions. This modification not only renders the model more adept at target identification but also renders it more lightweight and less computationally intensive. The optimal iteration of GhP2-YOLOm demonstrated exceptional accuracy in quantifying rapeseed flower samples, showcasing an impressive mean average precision at 50% intersection over union metric surpassing 95%. Leveraging the virtues of StrongSORT, the subsequent tracking of rapeseed flower buds and blossom patterns within the video dataset was adeptly realized. By selecting 20 video segments for comparative analysis between manual and automated counts of rapeseed flowers, buds, and the overall target count, a robust correlation was evidenced, with R-squared coefficients measuring 0.9719, 0.986, and 0.9753, respectively. Conclusively, a user-friendly “Rapeseed flower detection” system was developed utilizing a GUI and PyQt5 interface, facilitating the visualization of rapeseed flowers and buds. This system holds promising utility in field surveillance apparatus, enabling agriculturalists to monitor the developmental progress of rapeseed flowers in real time. This innovative study introduces automated tracking and tallying methodologies within video footage, positioning deep convolutional neural networks and multi-target tracking protocols as invaluable assets in the realms of botanical research and agricultural administration.

## 1. Introduction

*Brassica napus* (*B. napus*, AACC, 2n = 38) is an important ornamental plant and oil crop in the globalized economy [1,2,3]. For the past several decades, rapeseed’s versatile applications in the food industry, industrial production, and animal husbandry have made it highly valuable [4,5]. Primary components including rapeseed flower traits, such as quantity and crown size, rapeseed branching habits, and the overall morphology of rapeseed plants, provide valuable insights into the agronomic crop characteristics exhibited by rapeseed during its later growth stages [6,7,8]. These special traits can serve as indicators of rapeseed yield, which ultimately plays a pivotal role in shaping breeding strategies and may also offer farmers invaluable insights for devising hedging tactics that ensure crop stability and profitability.

The accurate counting of plants and plant organs holds significant phenotypic information for breeders and growers alike. The counting of plant organs, especially the number of flowers, reflects the transition of crops from vegetative growth to reproductive growth and improves better management of early yield for growers [9,10]. However, although manual counting is possibly the simplest method for counting seedlings and flowers, it has a few drawbacks, such as being a time-consuming procedure, being labor intensive, and having high potential for subjective errors, limitations that make it challenging to meet the demands of large-scale agricultural production [11,12,13]. As a consequence, developing an efficient, convenient, and cost-effective automatic counting technology for rapeseed flowers has become a pressing issue that urgently requires a solution in the current agricultural production landscape [9,14].

In contrast to traditional methods relying on manual feature extraction, which are task-specific, complex, and lack robustness, deep learning has emerged as a powerful tool in rapeseed phenotype recognition, fueled by advancements in hardware [10,15]. Zhao et al. merged backlight photography of rapeseed pods’ transparency with threshold segmentation, Faster R-CNN, and DeepLabV3+ to segment and count internal seeds in images acquired across different lighting intensities [16]. Wang et al. invented a method based on Detectron2 and Mask R-CNN so that depending on RGB image, people recognized and calculated phenotypes such as the length and width of rapeseed pods [17]. Shen et al. conducted a high-throughput phenotypic analysis of rapeseed plants based on Mask R-CNN and drone images [18]. Meanwhile, computer vision based on deep learning has also been widely applied in flower counting. For example, Tan et al. proposed an anchor-free deep learning model for counting seedlings and flowers, which utilizes a tracking-based counting approach that forgoes complex motion estimation algorithms, yielding counting results highly correlated with ground truth [19]. Bhattarai et al. presented a deep learning algorithm based on weakly supervised regression, offering a simplified approach for counting apple flowers and fruits without the need for precise detection and segmentation [20]. Estrada et al. used high-density images based on deep learning to detect and count the number of flowers in peach groves, which can serve as an early prediction indicator for crop yield [21]. Bi et al. proposed the ZGWO-RF model for non-destructive classification of 11 maize seeds based on multi-source data, which preliminarily explored a method for quantifying specific plant components in different genotypes [22]. The above studies illustrate that as a data-driven approach, deep learning automatically extracts high-level features from complex images, enabling superior performance and wider applicability in plant production tasks.

In recent years, there has been rapid advancement in the application of cameras in agriculture, revolutionizing modern farming practices. Field cameras, which are cost-effective and equipped with diverse sensors, have become invaluable tools for large-scale agricultural monitoring. These systems, which can be programmed quickly, enable real-time monitoring of field and crop conditions, facilitating the early detection and diagnosis of diseases and pests through data analysis. For instance, Oishi et al. constructed potato plant datasets from images and videos, and devised an automated anomaly detection system for potato plants utilizing portable cameras and advanced deep learning techniques [23]. Feng et al. presented a lightweight YOLOv7-based citrus detection and dynamic counting approach, tailored for modern planting scenarios in citrus image and video analysis [24]. Zhang et al. integrated prior knowledge from rice transplanter operations into a pixel-level classification of rice paddy regions and developed a multi-line detection method combining central axis-based point clustering with RANSAC [25]. These researches demonstrated that video technology had been applied in plant production. However, there has been no research combining video technology with machine vision and deep learning for the counting of rapeseed flowers.

In this study, we collected a rapeseed flower dataset using an iPhone 14, comprising RGB images and videos. The dataset was annotated using LabelImg 1.8.6 software, where both the RGB images and frames extracted from the videos were labeled. We utilized the YOLO v8 object detection algorithm as the baseline and made enhancements by incorporating the Ghost module and P2 detection head. These improvements aimed to enhance the accuracy of detecting small objects while maintaining a lightweight model. The enhanced GhP2-YOLOm model demonstrated outstanding performance on the rapeseed flower dataset, surpassing an mAP_50_ beyond 95% and achieving an F1-score of 0.880. Subsequently, by integrating the powerful StrongSORT multi-target tracking algorithm, real-time detection and counting of rapeseed flowers and blooms were effectively accomplished. Finally, we developed a visualization system based on GUI and PyQt5, which can be deployed on field detection equipment to enable farmers to monitor the growth of rapeseed flowers in the field at any time. This study showcases the application of deep learning in analyzing phenotype data of rapeseed flowers, providing valuable insights for industrial production. Furthermore, the research methods described in this publication can serve as a reference and guidance for the study of other crops.

## 2. Materials and Methods

### 2.1. Data Collection and Preprocessing

A total of ten videos, each lasting approximately 15 min, were recorded between 10:00 and 16:00 at the Farming Park of Zhejiang A&F University (119.72 E, 30.25 N) and the Plant Protection Station of Honggao Town, Yuhang District, Hangzhou City, Zhejiang Province, China. The varieties of rapeseed cultivation include “Zheda 630”, “Ruiyou 501”, “Chuanyou 45”, and “Qinyou 336”. The recording dates spanned from 5 to 25 March 2023, and from 15 to 20 March 2024. The images and videos were captured using two smartphones, an Apple iPhone 13 and Apple iPhone 14, under natural lighting conditions to document rapeseed flowers. During filming, the operator moved slowly through the fields at a speed of approximately 0.1 m per second. The smartphones were angled nearly 60° downward, and the shooting distance ranged from 10 to 30 cm from the rapeseed flowers. The dataset summary and details are presented in Table 1, while the experimental farming garden and rapeseed flowers are illustrated in Figure 1.

For the detection dataset, an image was extracted from every 15 frames of each video. These images, along with the photographs, were annotated using LabelImg software to create labels for rapeseed flowers. Due to the need for efficient training, images were randomly cropped to a size of 960 × 720 pixels from the captured photos and extracted frames. In total, 13,002 rapeseed flower buds and 11,679 blooming flowers were annotated across 1421 photos. Subsequently, the labeled images were divided into training, validation, and testing datasets in an 8:1:1 ratio for developing our rapeseed flower detection model.

### 2.2. Experimental Operating Environment

In this experimental setup, the operating system employed was Windows 11. The computational power was provided by a 12th Generation Intel(R) Core (TM) i5-12400F CPU clocked at 2.50 GHz, coupled with an NVIDIA GeForce RTX 3060 graphics card featuring 8 GB of dedicated memory. To establish the training environment, Anaconda3 was utilized, hosting Python 3.8, PyTorch 1.9.1, and torchvision 0.15.0 as the core programming and machine learning frameworks. To expedite the training process, NVIDIA CUDA 11.3 was leveraged for GPU acceleration. During the training phase, the input images were dynamically resized to a resolution of 640 × 640 pixels. LabelImg software was applied to manually label the rapeseed flower images as a training dataset. Among them, “1” represented the rapeseed flower bud, and “2” represented the blooming rapeseed flower (Figure 2).

### 2.3. Construction of Rapeseed Flower Dataset Depending on YOLO v8 Model Design

YOLO (You Only Look Once) is a one-stage object detection algorithm designed to recognize object classes and bounding boxes in images with a single pass [26,27,28]. YOLO v8, the latest iteration in the YOLO series developed by Ultralytics, boasts advantages such as lightweight architecture, strong scalability, and impressive detection performance. While YOLO v9 and YOLO v10 are scheduled for release in 2024, YOLO v8 remains the preferred choice for applications in agriculture and industry due to its superior performance [17,29].

In response to the challenges posed by complex backgrounds in rapeseed flower images and the detection of small targets such as flower bones, YOLO v8 has undergone enhancements and refinements. The introduction of the P2 structure in the network head aims to improve the detection of small objects, while the incorporation of Ghost modules in place of a few Conv modules in both the backbone and head strives for model lightweight and reduced parameter complexity. Simultaneously, some C2f modules have been replaced by C3Ghost modules. This enhanced model, known as GhP2-YOLO (Figure 3), delivers significantly improved detection accuracy, maintains model simplicity, and achieves a lightweight design. It presents an effective solution for automating crop identification in the field. Further technical details will be elaborated upon subsequently. The training parameters of the model are as follows (Table 2):

#### 2.3.1. Ghost Modules Introduction

The notion of the Ghost module stems directly from the foundational principles of GhostNet. In the context of the GhP2-YOLO architecture, the incorporation of GhostConv and C3Ghost components, both derived from the Ghost module, is a strategic move aimed at enhancing performance through a reduction in parameter count and computational intricacy [30,31]. A meticulous examination of the output feature maps from the initial residual group in ResNet-50 uncovers a remarkable similarity among them, hinting at a potential correlation between the robust feature extraction maps of convolutional neural networks (CNNs) and the emergence of these similar feature map pairs, colloquially known as Ghost pairs.

Contrasting with conventional approaches, the Ghost module embraces rather than evades the creation of Ghost pairs. It employs a strategy that leverages lightweight linear transformations to actively proliferate these Ghost pairs, thereby optimizing resource utilization. The traditional convolution module’s operational flow, as depicted in Figure 4A, undergoes a paradigm shift in the Ghost module, which elegantly partitions the process into three pivotal stages: initiating with conventional convolution for efficient initial feature extraction, then generating additional Ghost feature maps through simplistic linear manipulations, and, ultimately, fusing these components to enrich the overall feature representation. This innovative approach not only maintains the feature extraction capacity but also achieves a notable reduction in model complexity and computational demands.

Initially, assuming to the input data *X* ∈ R *^c×h×w^*, where *h* and *w*, respectively, represent the height and weight of the input data, and *c* is the number of input channels. The formula for the operation of a traditional convolutional layer used to generate n feature maps is as follows:*Y = X * f + b*

Certainly, *** represents the convolutional calculation and *b* is the bias term. The output feature maps are *Y* ∈ R*^h’×w’×n^* and the convolution filters in the layer are *f* ∈ R*^c×k×k×n^*. Meanwhile, f is the convolution filter, whose kernel size is *k × k*, and *h’* and *w’* are the height and width of the output data. Due to the fact that the number of filters *n* and the channel number *c* are both really enormous, the FLOPs value could be calculated as *n*·*h’*·*w’*·*c*·*k*·*k.*

In the context depicted in Figure 4, the feature map ensemble outputted by the convolutional layer exhibits pronounced redundancy, with discernible similarities observed among subsets of these maps. This underscores the fact that not every feature necessitates independent derivation through computationally intensive processes involving extensive floating-point operations and a profusion of parameters. Han et al. posit a compelling argument, contending that these redundant feature maps can be conceptualized as the augmented manifestations of a succinct set of underlying feature maps, achieved through streamlined transformations—termed “Ghost” mappings. These core feature maps, characterized by their parsimony in terms of both quantity and spatial dimensions, are fundamentally generated by fundamental convolutional kernels. Precisely, the primary convolution operation serves as the cornerstone, yielding m intrinsic feature maps, denoted as *Y’* ∈ R*^h’×w’×m^*, which subsequently serve as the efficient foundation for the derivation of the redundant feature maps:*Y’ = X * f’,*
where the applied filters are *f’* ∈ R*^c×k×k×m^*, m ≤ n, and for the purpose of simplicity, the bias term could be omitted. Other hyperparameters like the size of filters, padding, and stride are equal to the common convolution for maintaining consistency in the spatial size of the output feature map. By cheap linear operations to each intrinsic feature in *Y’*, *s* ghost features are generated as follows:$$y_{i,j}=\phi_{i,j}\left(y_{i}^{\prime }\right),\;\;\;\;\forall \;i=1,\;\ldots ,m,\;\;\;j=1,\;\ldots ,s,$$
where *y_i_^′^* in the formula means the *i*-th inherent feature map in *Y’*, $y_{i,j}$ is the *j*-th ghost feature map, and $\phi_{i,j}$ represents the *j*-th (except for the last) linear calculation in other words there can be at least one ghost feature maps ${\left\{y_{ij}\right\}}_{j=1}^{s}$ in *y_i_^′^*. As depicted in Figure 4, $\phi_{i,s}$ represents an identity mapping mechanism designed to maintain the inherent feature map integrity. The feature map *Y* = *[y*_11_*, y*_12_*, …, y_ms_]*, with dimensions *n = m · s*, serves as the output of the ghost module, achieving significantly reduced computational cost per channel compared to standard convolution operations.

Last but not least, implementing linear operations of uniform size within a Ghost module yields high efficiency in both the CPU and GPU due to practical optimizations. By concatenating all feature maps derived from the initial two steps, an acceleration notional ratio of upgrading the common convolution with the Ghost module can be calculated as follows:$$r_{s}=\frac{n\cdot h^{\prime }\cdot w^{\prime }\cdot c\cdot \;k\cdot k}{\frac{n}{s}\cdot \;h^{\prime }\cdot w^{\prime }\cdot c\cdot \;k\cdot k+\left(s-1\right)\cdot \;\frac{n}{s}{\cdot \;h}^{\prime }\cdot w^{\prime }\cdot d\cdot \;d\;}=\frac{c\cdot \;k\cdot k}{\frac{1}{s}\cdot \;c\cdot \;k\cdot k+\left(s-1\right)\cdot \;\frac{1}{s}\cdot \;d\cdot \;d\;}\;\approx \;\frac{s\;\cdot \;c}{s+c-1}\;\approx s,$$
where d × d is similar to k × k in magnitude and s << c. At the same time, the ratio can also be calculated:$$r_{c}=\frac{n\cdot c\cdot \;k\cdot k}{\frac{n}{s\;\;}\cdot c\cdot \;k\cdot k+\left(s-1\right)\cdot \;\frac{n}{s}\;\cdot d\cdot \;d\;}\;\approx \frac{s\;\cdot \;c}{s+c-1}\;\approx s,$$
which is equivalent to utilizing the speed-up ratio of the Ghost module.

In summary, GhostConv introduces a channel separation mechanism in Conv, which can divide a channel into several sub channels to obtain more information for free without consuming additional computing resources, ensuring that the number of parameters remains unchanged or even reduced.

#### 2.3.2. P2 Structure Applied in GhP2-YOLO Head

The conventional YOLOv8 architecture comprises three output layers—P3, P4, and P5—for object detection. To bolster its capacity to identify minute objects, we have augmented the model’s head section with an additional P2 layer [32]. This P2 layer incorporates expanded feature pyramid hierarchies, empowering it to precisely detect and localize smaller entities while efficiently fusing multi-scale information. Notably, the P2 layer minimizes convolutional operations and maintains a larger feature map dimension, fostering enhanced sensitivity towards small object recognition. Consequently, the model now boasts four output layers, preserving the Backbone’s output while refining the Neck and Head sections’ configurations.

The introduction of the P2 layer, specifically tailored for small target detection, significantly enhances YOLO v8’s proficiency in detecting objects that occupy minimal pixel space within images. Such targets, inherently prone to oversight or misclassification due to their diminutive presence, now benefit from the heightened sensitivity of the dedicated P2 layer. This advancement underscores YOLO v8’s ability to meticulously perceive and pinpoint small targets, thereby elevating the overall accuracy of small object detection.

### 2.4. StrongSORT Algorithm for Target Tracking

StrongSORT, an advanced multi-target tracking algorithm, is a refinement of DeepSORT that enhances tracking accuracy and overall performance through sophisticated improvement strategies [33]. Building on its predecessor SORT, StrongSORT integrates target detectors and trackers with a core utilizing Kalman filtering and the Hungarian algorithm (Figure 5). The former predicts the target’s position and state in future frames, while the latter optimizes the match between predicted and measured states. Despite performing well under unobstructed or continuous visibility conditions, the Kalman filter’s predictive ability may be limited when challenged by occlusion or target disappearance, resulting in tracking interruptions. DeepSORT addresses these challenges by leveraging deep learning models to extract target appearance features [34].

As an evolution of DeepSORT, StrongSORT refines appearance feature extraction by incorporating BoT (bag of tricks), employs an EMA (exponential moving average) to smooth feature updates for noise reduction, utilizes NSA (neural network-based appearance) for complex motion handling, and integrates MC (motion compensation) into the matching process to enhance accuracy. Additionally, the implementation of the ECC (external camera calibration) strategy corrects camera motion errors, and the adoption of woC (without cascading) simplifies the processing flow, leading to comprehensive enhancements in accuracy, robustness, and efficiency in multi-target tracking algorithms.

### 2.5. The Rapeseed Flower Detection Algorithm Development

As a Python graphical user interface (GUI) development library based on the Qt framework, PyQt5 provides a solid foundation for developers to build powerful and feature-rich applications with its cross-platform features and the efficiency of the Python language. This library integrates rich controls and components, including but not limited to dialog boxes, dropdown menus, toolbars, buttons, and text boxes, greatly facilitating the design and implementation of complex user interfaces. In addition, PyQt5 also has built-in support for multimedia elements such as images and sound, as well as flexible processing mechanisms for keyboard and mouse interaction, providing rich technical means to enhance user experience.

Given the above advantages, we designed and implemented a highly compatible and user-friendly GUI software using the PyQt5 module. This software aims to ensure stable operation on diverse hardware platforms, such as embedded devices like Raspberry Pi, and operating system environments like Windows and MacOS, demonstrating excellent cross-platform capabilities. The software interface layout is carefully planned and mainly divided into two visual areas: one area is used to visually display the comparison images before and after image processing, and the other area records and displays operation logs and related information in real-time in text form for user monitoring and analysis.

It is particularly worth mentioning that the software interface cleverly integrates two control buttons, which are, respectively, used to start and stop the capture and processing flow of video streams. Users can trigger the automatic connection of the video stream to the camera or designated video source of the field observation equipment with a simple click operation and set the system to automatically perform image prediction tasks every 0.5 s. The predicted results will be instantly fed back to the user interface, and the related log information will also be updated synchronously to the text log area, ensuring that users can obtain comprehensive and accurate feedback information in real time, greatly improving the practicality and user satisfaction of the software. The interface of the real-time detection software for rapeseed flowers designed based on GUI and PyQt5 is shown in Figure 6.

### 2.6. Model Training

In our endeavor to optimize object detection and tracking models on the rapeseed flower dataset and enhance their generalization capability, a systematic approach was adopted. The dataset was methodically split into training, validation, and testing sets in an 8:1:1 ratio, allowing for effective evaluation and adjustment of the model at different stages. To enhance model stability and reliability, ten independent training cycles were implemented, each employing five distinct random seeds for dataset shuffling to explore the impact of data diversity on model learning. Setting the epoch parameter to 150 during training ensured sufficient iterations for comprehensive data feature learning.

During the model construction phase, transfer learning (TL) technology was skillfully integrated, leveraging cross-domain knowledge transfer to expedite and optimize the learning process for the target task. Initially, leveraging an Ultralytics model pre-trained on the extensively annotated COCO dataset provided a foundation rich in visual priors. Subsequently, parameters from this pre-trained model were transferred to a new Ghost-based model tailored for the rapeseed flower dataset. Shared base layer parameters from the pre-trained model were retained for their robust feature extraction capabilities, while unique new model layers had parameters initialized randomly to address specific rapeseed flower dataset requirements. This strategy aimed to merge advanced Ghost features with TL efficiency advantages, significantly enhancing model performance in rapeseed flower detection tasks.

Upon completion of training the object detection model, the StrongSORT algorithm was integrated to achieve precise tracking of blooming rapeseed flowers and flower buds. By correlating detection targets with existing trajectories, continuous and stable tracking of target object motion states was achieved, ultimately producing an image sequence with tracking information.

## 3. Results

### 3.1. The Rapeseed Flower Detection Result Based on GhP2-YOLO

Firstly, three configurations of YOLO v8 (n, s, and m) were trained and tested for detecting rapeseed flower buds and blooming rapeseed flowers. To address the dataset’s limited image count, various data augmentation techniques were employed, including flipping, rotating, cropping, adding noise, blurring, masking, color conversion, and other methods, aimed at enhancing model generalization capability and mitigating potential sample imbalance issues. The dataset was partitioned into training, testing, and validation sets in an 8:1:1 ratio. The training process spanned 150 epochs with a learning rate of 0.01, weight decay of 0.005, and momentum of 0.937, utilizing optimizers such as SGD, Adam, and AdamW. Among these models, YOLO v8m achieved the highest mAP_50_ value of 0.916, with mAP_50–95_ at 0.755. A detailed breakdown of precision, recall, F1, mAP_50_, and mAP_50–95_ values for each object in the rapeseed flower dataset revealed recognition accuracies of 0.825 for rapeseed flowers and 0.873 for flower buds. Their mAP_50_ values exceeded 90%, with mAP_50–95_ around 0.75 as illustrated in Table 3.

In the enhanced GhP2-YOLO model, YOLO v8n, YOLO v8s, and YOLO v8m models of three different sizes were utilized, with epochs set at 150 and AdamW optimizer. Post-data augmentation, the trained images displayed improved evaluation metrics compared to the baseline. In GhP2-YOLOm, the recognition accuracy for rapeseed flower buds reached 0.913, the recall rate was 0.894, and mAP50 for both rapeseed blooming flowers and buds surpassed 95%. For GhP2-YOLOn, the mAP50 values for the two categories were 0.893 and 0.929, respectively.

### 3.2. GhP2-YOLO Ablation Experiment

In this deep learning object detection research using the rapeseed flower dataset, we initially conducted baseline experiments based on YOLO v8n, YOLO v8s, and YOLO v8m models. The results showed that YOLO v8m achieved commendable performance in key indicators such as precision (0.851), recall (0.764), mAP_50_ (0.916), mAP_50–95_ (0.755), and F1-score (0.851). To further enhance the model, we increased the number of layers and adopted the YOLO v8m variant. This improvement significantly increased recall, mAP50, mAP_50–95_, and F1-score, while precision remained relatively stable.

In order to further improve the performance of the model, especially for small object detection tasks, and make the model lighter, we introduced the Ghost module in the model, replacing some convolution layers with GhostConv and some C2f modules with C3Ghost modules, and added P2 structure in the header structure. Through ablation experiments, it was verified that these structural improvements not only increased the complexity of the model but also significantly improved all five evaluation metrics.

It is obvious that the introduction of the P2 structure has a greater promoting effect on the increase in accuracy indicators, while the introduction of the Ghost module is more beneficial in reducing the complexity and computational complexity of the model. Specifically, in the GhP2-YOLO model that integrates Ghost and P2 structures, precision increased to a maximum value of 0.889, recall reached 0.871, mAP_50_ and mAP_50–95_ increased to 0.955 and 0.782, respectively, while F1-score also reached the highest value of 0.880. These improved indicators indicate that the YOLO v8 model, which has been structurally optimized, performs well in detecting small target objects such as rapeseed flower buds and has high recognition accuracy (Table 4).

### 3.3. Model Comparison

To further validate the performance of the GhP2-YOLO model, this study conducted an extensive comparison with multiple classic one-stage and two-stage object detection models, evaluating metrics such as mAP_50_, mAP_50–95_, parameters, and FLOPs, as detailed in Table 5. The YOLO v8 series consistently outperforms other first-stage deep learning models like YOLO v5s and YOLO v7-tiny, as well as outperforming second-stage models like Faster R-CNN. This superiority is attributed to the task alignment allocator matching technology employed in YOLO v8, which diverges from traditional IoU matching and one-sided proportional allocation methods. YOLO v8 adopts a C2f structure with enhanced gradient flow to adjust channel numbers for different scale models. In the head section, the past coupled head structure is replaced by the prevalent decoupled head structure, segregating the classification and detection components, and transitioning from Anchor-Based to Anchor-Free for improved speed and accuracy in small object detection.

In the pursuit of enhancing contextual information learning, the model captures supervised signals concerning relationships and directly conveys them to the feature map. Regarding detection accuracy, GhP2-YOLO surpasses the other nine assessed models, with YOLO v8m emerging as the second-best performer following the GhP2-YOLO series. This suggests that the YOLO series excels in detection performance within classical object detection frameworks. Renowned for their compact size and high recognition accuracy, YOLO-series models benefit significantly from mosaic data augmentation, which balances the distribution of large, medium, and small targets, thereby bolstering model robustness.

Bubble plots were plotted using mAP_50_, mAP_50–95_, and FLOPs from 10 different models (Figure 7). Although the parameter count and gradient of the GhP2-YOLO series models are smaller compared to their baseline, the FLOPs of GhP2-YOLOn and GhP2-YOLOs are larger than their baseline models YOLO v8n and YOLO v8s, which means that the algorithm requires more floating-point operations to complete their computational tasks. Under certain hardware resource constraints, larger FLOPs may consume more computing resources, resulting in high energy consumption and increased hardware costs.

Surprisingly, the FLOPs of the GhP2-YOLOm model are smaller than its baseline model YOLO v8m. Considering that GhP2-YOLOm is also the best-performing model in rapeseed flower detection, it is applied in the StrongSORT multi-target tracking task in the following text and used as a parameter for developing visualization pages based on GUI and PyQt5.

### 3.4. Multi-Object Rapeseed Flower Tracking Results Based on StrongSORT

To evaluate the matching process of GhP2-YOLO with the StrongSORT algorithm and the effectiveness of the two-point enhancement of the Kalman filter on multi-target tracking of rapeseed flowers, the rapeseed flower tracking dataset established in this study was utilized to train and test the StrongSORT algorithm based on the previously obtained results from GhP2-YOLOm (Figure 7). A comparative analysis was performed against the current multi-target tracking algorithm DeepSORT, which also adopts the deep appearance continuity paradigm. The results are documented in Table 6. The comparison reveals that in comparison with DeepSORT, StrongSORT showcased a notable increase in the MOTA index by 4.9%, HOTA by 2.8%, and IDF1 by 7.8%. This enhancement can be attributed to the refined trajectory and target matching process in StrongSORT, incorporating both appearance and motion characteristics, alongside advancements in the matching process and Kalman filtering.

While DeepSORT also leverages appearance and motion characteristics, its approach to appearance feature extraction is relatively straightforward, with motion features primarily utilized to eliminate implausible matches, leading to excessive reliance on appearance features. Moreover, the similar appearance between rapeseed flowers and flower buds can exacerbate identity-switching issues. Notably, all multi-target tracking algorithms in this study employ GhP2-YOLO as the detector, resulting in minimal differentiation between the two algorithms in terms of MOTP metrics. DeepSORT and StrongSORT exhibit lower MOTP metrics due to the target box output of DeepSORT after the Kalman filter update stage, impacting the box position based on Kalman filtering adjustments.

### 3.5. Linear Regression Analysis between Manual and Machine Counting

The proposed method for counting rapeseed flowers and flower buds was tested by evaluating 20 video clips cut from the counting dataset. The number of rapeseed flowers and flower buds in each video clip was manually counted three times, and the average of the three values was taken as the final quantity. A simple linear regression analysis was conducted on the machine counting and manual counting of rapeseed flower buds, flowers, and total target numbers, as presented in Figure 8 and Table 7.

Table 7 presents the comprehensive statistical data comparing the actual measurement and machine counting of rapeseed, encompassing the sum of squared errors (SSE), mean squared error (MSE), root mean squared error (RMSE), and the coefficient of determination (R²). Figure 8A displays the results of the machine manual technique for rapeseed flower buds: R^2^ = 0.986, y = 0.9619x + 2.8265. In Figure 8B, the R^2^ for machine manual counting of rapeseed flowers is 0.9719, and the linear regression function is y = 0.9998x−1.0354. Figure 8C demonstrates that the counting algorithm for total objects in the rapeseed flower dataset yielded highly similar results to manual counting (R^2^ = 0.9753), with a linear regression function of y = 0.9927x + 0.3606. Drawing upon the data presented in Table 7 and Figure 8, a conclusion can be drawn that there exists a strong correlation between the manually and machine-measured counts of rapeseed flower buds, flowers, and the total number. This observation underscores the feasibility of employing deep learning models for the purpose of accurately counting rapeseed flower buds and flowers. It can be observed that the counting results of rapeseed flower buds are more accurate than those of rapeseed flowers. This could be attributed to the more uniform shape of rapeseed flower buds and the specific improvements in GhP2-YOLO for small target, low-resolution image recognition, facilitating easier machine recognition of rapeseed flower buds. Conversely, rapeseed flowers exhibit a more diverse morphology, intertwined backgrounds, and greater complexity, resulting in relatively lower accuracy in counting and recognition.

For all test videos, the absolute counting error of 95% of rapeseed flower bud test results is less than or equal to 3, and the absolute counting error of 90% of rapeseed flower test results is less than or equal to 5. Additionally, Figure 8D illustrates that 85% of the total object counting results of rapeseed flowers have an absolute error of less than or equal to 5, indicating acceptable counting accuracy. These findings suggest the potential for utilizing the results of this method directly for counting rapeseed flowers and flower buds.

### 3.6. Visualization of Rapeseed Flower Detection

A visualization system called “Rapeseed flower detection” developed by GUI and PyQt5 utilizes a camera module connected to a field detection system to continuously capture video clips of rapeseed flowers. The video is then processed in real time to detect the number of rapeseed flowers and rapeseed flower buds. The system combines the improved GhP2-YOLOm detection algorithm with the StrongSORT multi-target detection algorithm, employing advanced deep learning techniques to accurately identify and locate rapeseed flowers and rapeseed flower buds within the captured frames. The system highlights the detected rapeseed flowers and buds with bounding boxes on the video feed, facilitating users in locating target objects effectively. Real-time counting of rapeseed flowers and flower buds are marked in the resulting image (Figure 9).

## 4. Discussion

Flowers play a crucial role as an essential organ for plants, bearing responsibilities such as pollination and fertilization, ensuring population continuity, and influencing crop yield and quality. Their vibrant blooms attract insect vectors for pollination while efficiently accumulating nutrients through photosynthesis, laying a solid foundation for plant growth and development [35]. During the flowering phase, precise quantification of flora and their anatomical structures is paramount for botanical breeders and agricultural cultivators, as it informs selective breeding programs, enhances crop resilience, and optimizes yield potential by understanding the intricacies of plant reproduction and growth. Despite the significance of flower traits, traditional manual counting methods for rapeseed flowers are labor-intensive and time-consuming. To address this issue, a novel counting and tracking method for rapeseed flowers and buds was developed in this study, utilizing deep learning and multi-objective tracking algorithms to achieve high-throughput and efficient detection.

A rapeseed flower dataset was curated using RGB images and videos for this study. The enhancement in detection accuracy for small targets was achieved in YOLO v8 by incorporating the Ghost module and P2 structure. Replacing certain convolution and C2f modules with lighter GhostConv modules and C3Ghost modules, a refined model named GhP2-YOLOm (an improvement of YOLO v8m) was selected after comparative experiments with classic deep learning object detection models. GhP2-YOLOm demonstrated promising performance on the rapeseed flower dataset. It is worth mentioning that GhP2-YOLOm was tested on the rapeseed flower dataset, and its precision and recall reached 0.889 and 0.871, respectively. GhP2-YOLOm’s F1-score also reached 0.880, achieving mAP_50_ values of 0.955 and mAP_50–95_ values of 0.782. Notably, with a parameter count of only 14.69M, GhP2-YOLOm excelled in detecting rapeseed flowers and buds. Furthermore, integrating the StrongSORT multi-target detection algorithm enabled real-time tracking and recognition of flowers and flower buds in rapeseed videos. A user-friendly rapeseed flower recognition system was established, leveraging a GUI and PyQt5 interface for visualizing detection outcomes. Our endeavors primarily revolve around the seminal exploration of harnessing deep learning paradigms for the recognition and enumeration of rapeseed blossoms, coupled with the development of an intricate visualization system geared towards real-time monitoring. Our research endeavors are meticulously crafted with an eye towards future applications, specifically targeting embedded devices like the Raspberry Pi. As we progress, our focus will intensify on refining and optimizing these solutions, ensuring their seamless integration and efficacy in practical scenarios. The system is envisioned to be deployed in field detection equipment, empowering real-time monitoring of rapeseed flower growth processes and data collection through field surveillance cameras.

StrongSORT is chosen as the target tracking method because it excels in multi-target tracking by combining advanced deep learning feature extraction with an efficient online tracking-detection framework. It effectively handles complex scenarios, such as target occlusion, fast movements, and interactions between multiple targets, ensuring stability and accuracy. For applications requiring both real-time performance and high precision, StrongSORT is the ideal choice, seamlessly integrating deep features with advanced tracking algorithms for smooth and reliable multi-target tracking. The StrongSORT was chosen for target tracking in rapeseed flower detection due to its excellence in multi-target tracking. It combines advanced deep learning feature extraction with an efficient online tracking detection framework, making it ideal for handling fast movements and interactions between multiple flower targets in the video. StrongSORT ensures stability and accuracy, making it the preferred choice for applications demanding both real-time performance and high precision.

Unfortunately, there are several limitations in our research. While using the MOT method to count agricultural objects from video frames has the potential to address occlusion issues compared to static image detection, the accuracy of object detection has a significant impact on counting accuracy. Continuous false negatives can lead to underestimation, while false positives can result in overestimation. Moreover, objects that are correctly detected in certain frames may disappear or go undetected in subsequent frames due to occlusion or specular reflection, and then reappear later and be correctly detected. In such cases, a new tracker is assigned to the reappearing object, leading to an overestimation of the final count. To overcome these issues, future improvements can be made by exploring different multi-target tracking algorithms and associating the same objects through appearance embedding to address the problem of duplicate counting due to occlusion and reappearance.

Occlusion is another challenge that affects detection accuracy. The moving camera setup may cause certain targets to only appear in certain frames. Additionally, the raceme infinite inflorescence of rapeseed flowers means that some flowers may be hidden behind others, and as the camera moves, some flowers may be partially or completely obscured by leaves. This occlusion can result in insufficient features for the detector, leading to false negative detections. Collecting more occluded rapeseed flower images to train the network can improve the detector’s accuracy for occluded objects. Additionally, instance segmentation techniques offer a refined approach to precisely delineate occluded flowers and discern their backgrounds, thereby enhancing the accuracy of identification. Unmanned aerial vehicle (UAV) and hyperspectral technology can also be used to detect rapeseed flowers. Nevertheless, the implementation of such methods necessitates a substantial investment in both human labor and material resources, including the annotation of large datasets by experts, the procurement of high-performance computational infrastructure, and the development of sophisticated algorithms capable of handling complex occlusions and background variations. 

Moreover, the lighting conditions and camera exposure during data collection on-site can impact image quality. Despite using an iPhone 14 with realistic shooting effects, some collected images were overexposed or had blurred backgrounds, making it difficult to distinguish yellow flowers from the background and resulting in false detections. Meanwhile, factors such as time of day, rain, sunlight, fog, and humidity also have an impact on the accuracy of image collection. However, due to device limitations, we only used mobile phones to collect data, which may limit the applicability of this solution to specific hardware such as cameras. It is important to emphasize the need for more universal solutions by testing the performance of the model on different exposure parameters, including using other smartphones, capturing images, or obtaining training data from various devices. Otherwise, the high accuracy of model parameters trained on a single dataset may be due to overfitting of the model. Therefore, improving photography conditions, enriching dataset sources such as collecting data in different external environments, and utilizing multiple devices including different models of mobile phones and cameras can help overcome these challenges.

This research significantly facilitates mechanized and intelligent field management, thereby reducing human labor costs and significantly enhancing farmers’ planting efficiency and productivity. By embracing advanced technological solutions like deep learning, it paves the way for more streamlined agricultural operations and ultimately boosts the profitability of farming endeavors.

## 5. Conclusions

To sum up, this research endeavors to enhance the accuracy and efficiency of rapeseed flower and flower bud counting by integrating a novel deep learning-based multi-target tracking approach. Specifically, this study introduces modifications to the YOLO v8 model through the incorporation of a P2 structure and Ghost module, resulting in the GhP2-YOLOm architecture. This tailored model demonstrates superior performance in detecting flowers and flower buds compared to the original YOLO v8 backbone, characterized by a reduction in model parameters and computational demands while maintaining or even improving detection accuracy. Notably, the P2 detection head plays a pivotal role in mitigating the challenges posed by scale variance, particularly in detecting small targets, by leveraging underlying features more effectively.

Furthermore, the integration of the StrongSORT model into the framework enables seamless tracking and counting of rapeseed flowers and buds, eliminating the necessity for intricate motion estimation mechanisms. The resulting visualization system, designed for rapeseed flower detection, offers unparalleled convenience for field personnel, facilitating real-time monitoring and assessment of floral growth. Ultimately, the GhP2-YOLOm model, with its capacity for automated tracking and counting of plant flowers and buds from video footage, holds significant promise for enhancing replanting strategies and advancing plant breeding efforts.

## Figures and Tables

**Figure 1 plants-13-02388-f001:**
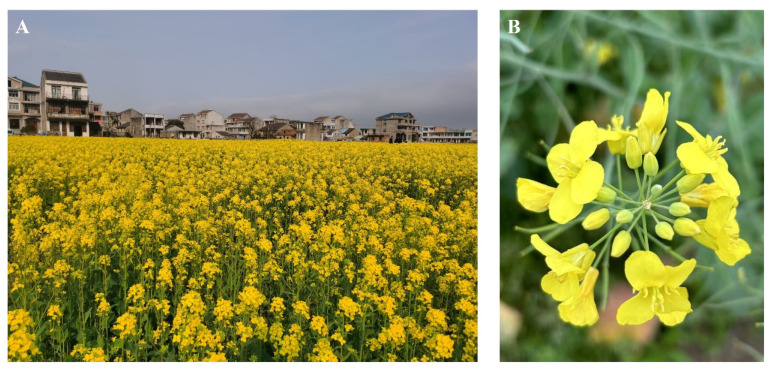
Rapeseed flowers in fields: (**A**) Experimental field garden. (**B**) Sample image of shooting rapeseed flowers after preprocessing.

**Figure 2 plants-13-02388-f002:**
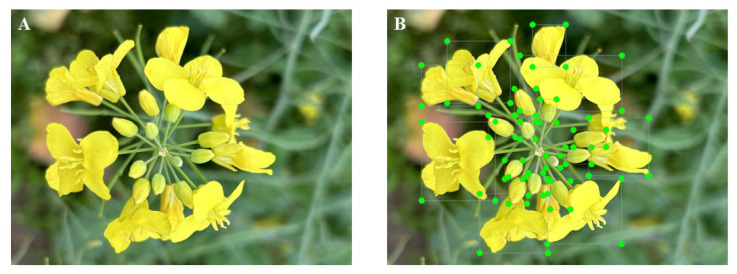
Example diagram of rapeseed flower annotation: (**A**) Original image. (**B**) Annotated image. (**C**) LabeImg 1.8.6 software annotation page example image.

**Figure 3 plants-13-02388-f003:**
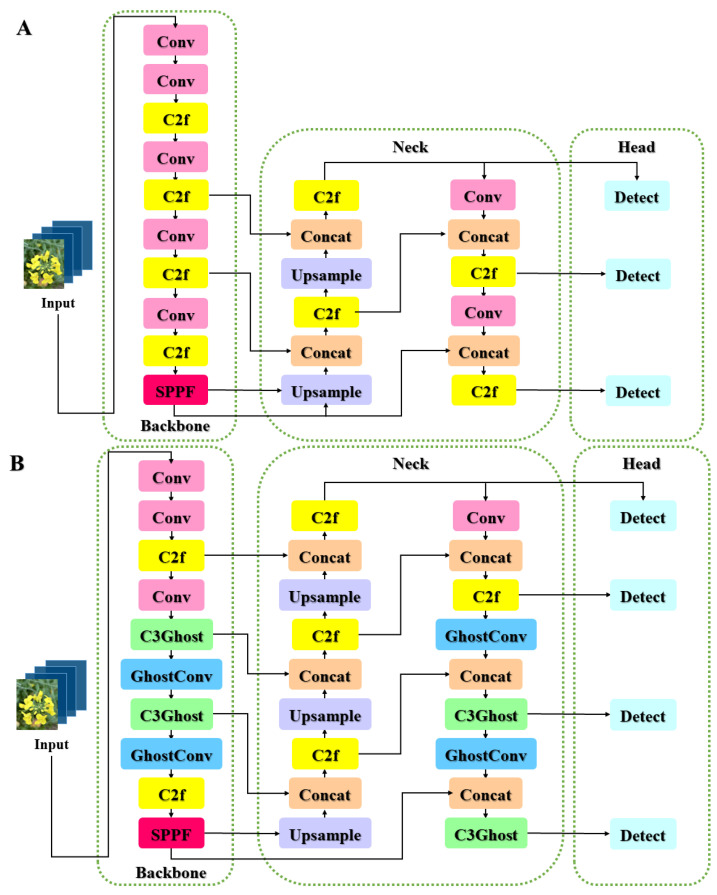
Comparison of YOLO v8 model structure before and after improvement: (**A**) YOLO v8 model structure. (**B**) GhP2-YOLO model structure.

**Figure 4 plants-13-02388-f004:**
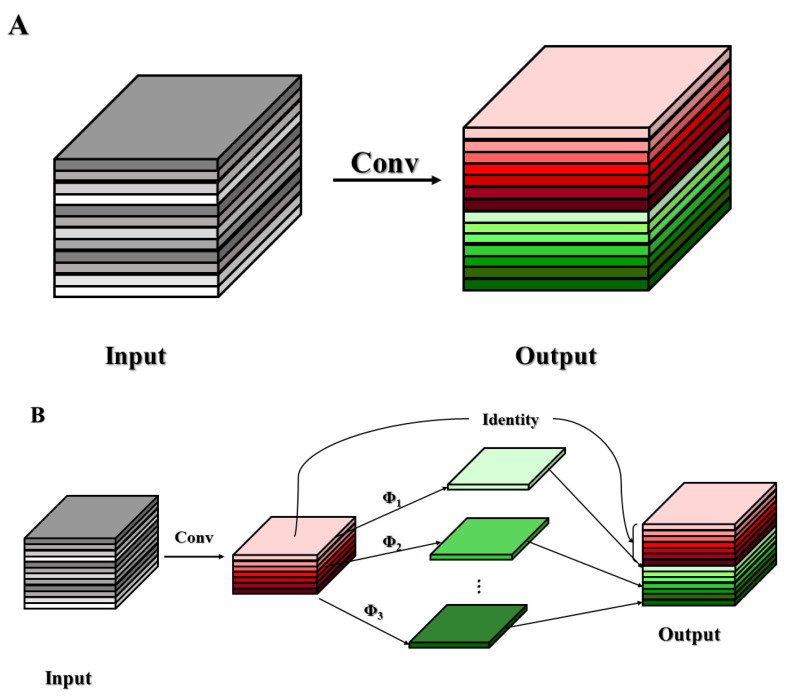
The conventional convolutional layer and Ghost module are used to output an example graph of the same number of feature maps: (**A**) Conventional convolution module. (**B**) Ghost module.

**Figure 5 plants-13-02388-f005:**
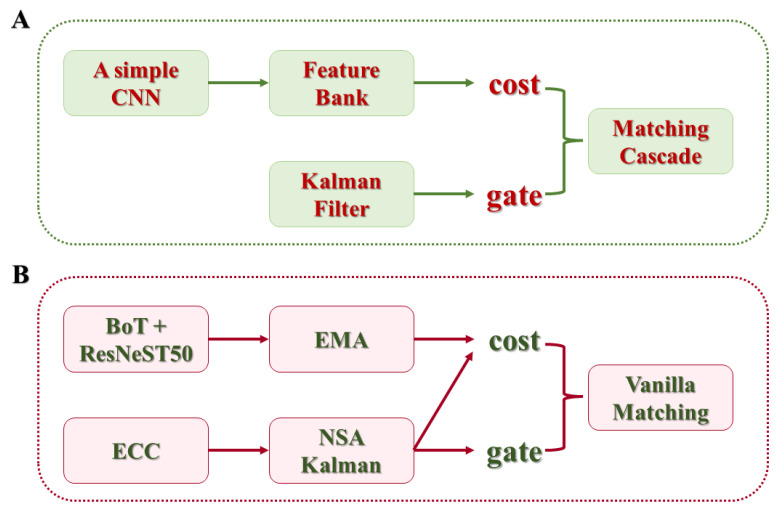
Comparison of DeepSORT and StrongSORT frameworks: (**A**) DeepSORT. (**B**) StrongSORT.

**Figure 6 plants-13-02388-f006:**
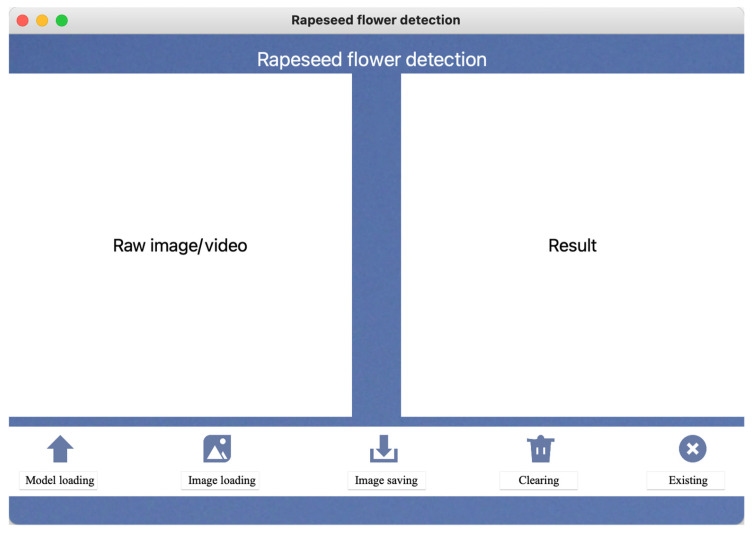
GUI visualization interface diagram based on Python and PyQt5 development.

**Figure 7 plants-13-02388-f007:**
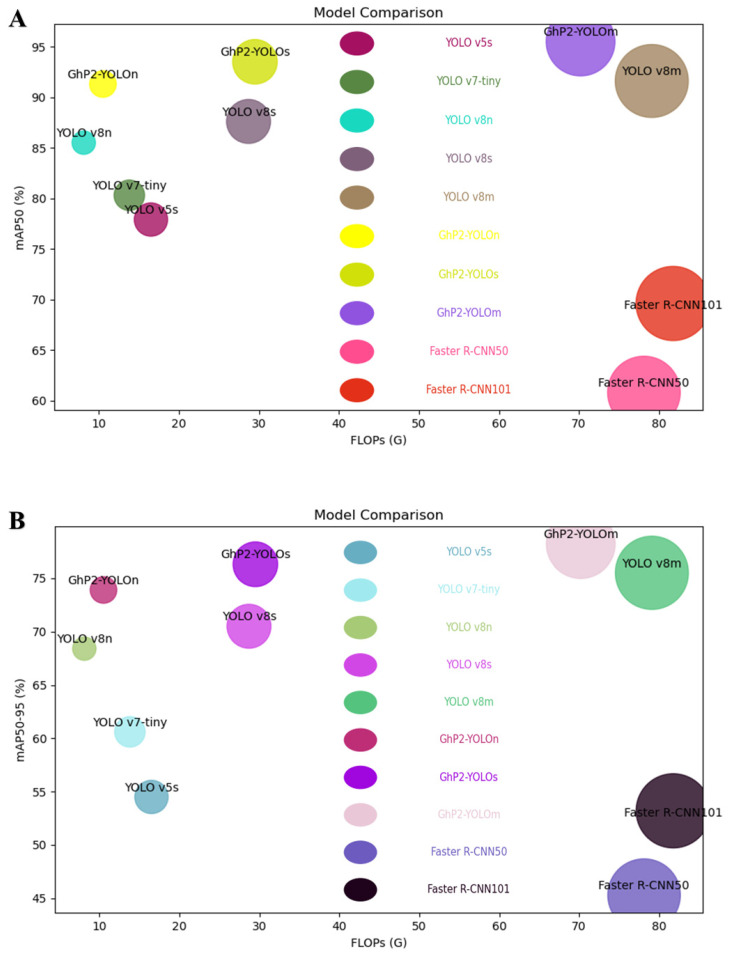
Comparison of detection performance: (**A**) The relationship between mAP_50–95_ and FLOPs (G). (**B**) The relationship between mAP_50_ and FLOPs (G).

**Figure 8 plants-13-02388-f008:**
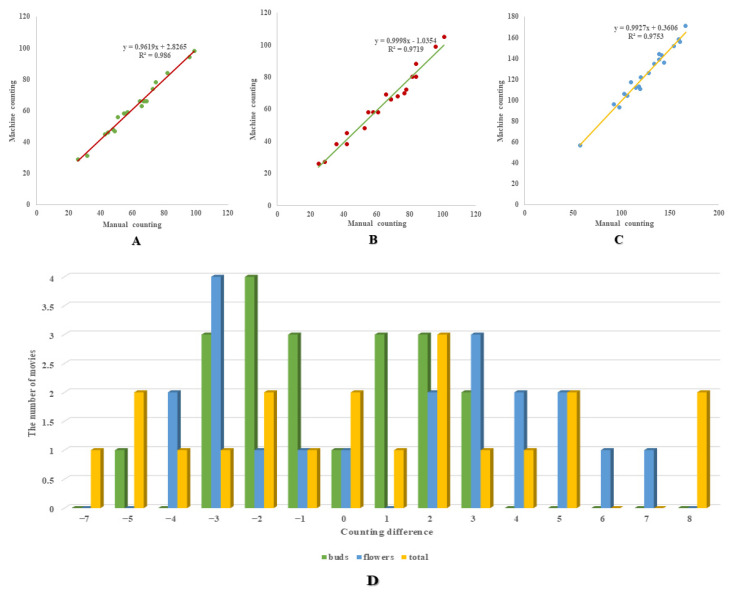
Comparison results of the manual and machine counting linear regression analyses of (**A**) rapeseed flower buds, (**B**) rapeseed flowers, and (**C**) total objects. (**D**) Bar chart between the number of movies and counting difference.

**Figure 9 plants-13-02388-f009:**
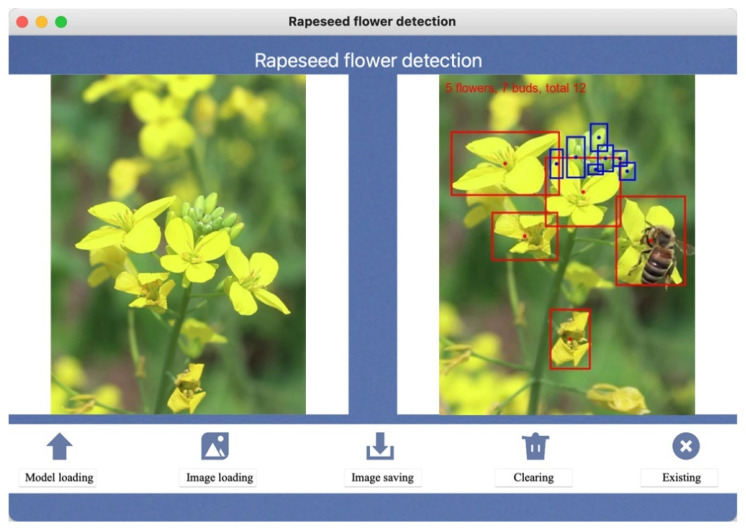
System visualization result page display based on GUI and PyQt5 development.

**Table 1 plants-13-02388-t001:** Dataset description.

Definition	Value
Number of video shoots	20
Total video duration	30 min
Video resolution	3024 × 4023 pixels
The total number of frames extracted from the video	1200
Total number of captured images	221
Image resolution	3024 × 4023 or 1280 × 1707 pixels

**Table 2 plants-13-02388-t002:** A few hyperparameters set in YOLO v8 models’ training.

Model	Epoch	Batch Size	Learning Rate	Weight Decay	Optimizer	Momentum	Parameters	Gradients	FLOPs(B)
YOLO v8n	150	16	0.01	0.0005	SGD	0.937	3,012,213	3,012,197	8.1 G
YOLO v8s					Adam		11,138,309	11,138,293	28.7 G
YOLO v8m					AdamW		25,860,373	25,860,357	79.1 G
GhP2-YOLOn					AdamW		2,065,980	2,065,964	10.5 G
GhP2-YOLOs					AdamW		7,164,748	7,164,732	29.5 G
GhP2-YOLOm					AdamW		14,687,564	14,687,548	70.2 G

**Table 3 plants-13-02388-t003:** The detection result of rapeseed blooming flowers and buds by different YOLO v8 models.

Model	Object Name	Precision	Recall	F1	mAP_50_	mAP_50–95_
YOLO v8n	Blooming flowers	0.782	0.756	0.769	0.844	0.669
	Flower buds	0.828	0.654	0.731	0.865	0.697
YOLO v8s	Blooming flowers	0.794	0.778	0.786	0.871	0.686
	Flower buds	0.845	0.698	0.764	0.881	0.721
YOLO v8m	Blooming flowers	0.825	0.799	0.812	0.906	0.747
	Flower buds	0.873	0.734	0.797	0.925	0.761
GhP2-YOLOn	Blooming flowers	0.819	0.803	0.811	0.893	0.735
	Flower buds	0.868	0.851	0.859	0.929	0.743
GhP2-YOLOs	Blooming flowers	0.832	0.829	0.830	0.921	0.745
	Flower buds	0.892	0.872	0.882	0.946	0.779
GhP2-YOLOm	Blooming flowers	0.861	0.844	0.852	0.951	0.766
	Flower buds	0.913	0.894	0.903	0.959	0.795

**Table 4 plants-13-02388-t004:** The ablation experiment resulted in different improved structures and models.

Model	Improved Structure	Precision	Recall	F1	mAP_50_	mAP_50–95_
YOLO v8n	/	0.807	0.701	0.750	0.855	0.684
	P2	0.823	0.776	0.799	0.901	0.725
	Ghost	0.819	0.731	0.773	0.877	0.692
	P2 + Ghost	0.846	0.829	0.837	0.913	0.739
YOLO v8s	/	0.822	0.735	0.776	0.876	0.705
	P2	0.867	0.836	0.851	0.929	0.758
	Ghost	0.839	0.782	0.809	0.896	0.714
	P2 + Ghost	0.865	0.852	0.858	0.935	0.763
YOLO v8m	/	0.851	0.764	0.805	0.916	0.755
	P2	0.886	0.869	0.877	0.946	0.779
	Ghost	0.864	0.811	0.837	0.921	0.753
	P2 + Ghost	0.889	0.871	0.880	0.955	0.782

**Table 5 plants-13-02388-t005:** The detection results of comparison models.

Category	Model	Backbone	Image Size	mAP_50_	mAP_50–95_	Parameters	FLOPs
One-stage	YOLO v5s	CSP-Darknet53	640 × 640	0.779	0.545	7.21 M	16.5 G
	YOLO v7-tiny	TinyDarknet		0.803	0.606	6.22 M	13.8 G
	YOLO v8n	C2f-CSP-Darknet53 C2f-CSP-Darknet53 C2f-CSP-Darknet53		0.855	0.684	3.01 M	8.1 G
	YOLO v8s			0.876	0.705	11.13 M	28.7 G
	YOLO v8m			0.916	0.755	25.86 M	79.1 G
	GhP2-YOLOn	Ghost-P2-YOLO v8		0.913	0.739	2.07 M	10.5 G
	GhP2-YOLOs			0.935	0.763	7.16 M	29.5 G
	GhP2-YOLOm			0.955	0.782	14.69 M	70.2 G
Two-stage	Faster-RCNN	Resnet50		0.608	0.453	41.12 M	78.12 G
	Faster-RCNN	Resnet101		0.696	0.532	61.16 M	81.77 G

**Table 6 plants-13-02388-t006:** Metrics of different multi-object tracking algorithms for tracking test videos.

MOT Algorithm	MOTA/%	HOTA/%	IDF1/%
DeepSORT	64.5	54.3	66.5
StrongSORT	69.4	57.1	74.3

**Table 7 plants-13-02388-t007:** Several statistical data on rapeseed flower buds, flowers, and total.

	SSE	MSE	RMSE	R^2^
Buds	104	5.2	2.280351	0.986
Flowers	275	13.75	3.708099	0.9719
Total	349	17.45	4.17732	0.9753

## Data Availability

The raw data supporting the conclusions of this article will be made available by the authors on request. The data used in our research are not publicly available, as they are also being utilized in an ongoing study.

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
