# Peer review of "Rapeseed Flower Counting Method Based on GhP2-YOLO and StrongSORT Algorithm"

_plants, 2024, doi:10.3390/plants13172388_

Round 1

Reviewer 1 Report

Comments and Suggestions for Authors

The article titled "A Rapeseed Flower Counting Method Based on GhP2-YOLO and StrongSORT Algorithm" presents a method for counting rapeseed flowers using image acquisition performed via a smartphone, followed by deep learning algorithms (GhP2-YOLO) and the StrongSORT tracking method. The main objective of the article is to highlight the possibility of improving accuracy in detecting small objects (flowers). The proposed method aims to replace manual flower counting, a process important for crop management.

Overall, I consider the article to be well-conceived, constructed, and presented. However, several points should be emphasized in the text:

·      Limited source dataset size: The dataset, consisting of 20 video clips and 1,421 images captured using a single device, may limit the applicability of the solution to that specific hardware (camera). It is important to stress the need for a more generalized solution by testing the model's performance on images captured with different exposure parameters (including using other smartphones) or acquiring training data from various devices. The high precision achieved might result from overfitting the model to a homogeneous dataset, which could be a case of cherry-picking.

·      Lack of reference to image acquisition conditions: The article only specifies the date range during which the images were captured. It would be valuable to include a discussion of how factors like time of day, rain, sunlight, fog, humidity, etc., could affect the accuracy of the measurements. This element should be added to the discussion to address potential external environmental influences.

·      Insufficient discussion of occlusion issues: The problem of occlusion or obstruction is only briefly mentioned. In practice, this is an important challenge that the model needs to handle. This should be further discussed, explaining how the method addresses situations where flowers are partially or completely hidden from the camera.

·      Why StrongSORT was chosen: While the StrongSORT method for object tracking is described, it would be beneficial to explain why this particular method was chosen. Were any tests performed using other tracking methods? A comparison with alternative techniques would strengthen the justification for using StrongSORT.

·      Field application of the GUI: Although the article touches on the topic of the application's interface, it does not sufficiently address its utility in field conditions. How well does the system function in agricultural environments? If everything is processed in post-processing, this should be clearly stated. Additionally, the equipment used (GPU RTX 3060) does not seem to be practical for widespread use in (agriculture fieldwork). Further clarification of the practical utility of the solution in real-world agricultural settings is needed.

Positive Aspects of the Article:

Despite the mentioned concerns, I would like to highlight several positive aspects of the article:

·      The article presents a promising automated method for flower counting, which has significant potential to improve the efficiency of crop management.

·      According to the results obtained and presented by the authors, the accuracy of the method is very high. However, attention should be given to the potential issue of model overfitting.

·      The article demonstrates a good reduction in the base CNN model's complexity through the application of the Ghost module, making the approach more computationally efficient.

·      There is potential for extending the method to other types of flower clusters and additional agricultural applications, suggesting a broad applicability for the proposed system.

Therefore, after introducing a few revisions in the text, I consider the reviewed article worthy of publication.

Author Response

From the Reviewer #1:

Comments and Suggestions for Authors:

The article titled "A Rapeseed Flower Counting Method Based on GhP2-YOLO and StrongSORT Algorithm" presents a method for counting rapeseed flowers using image acquisition performed via a smartphone, followed by deep learning algorithms (GhP2-YOLO) and the StrongSORT tracking method. The main objective of the article is to highlight the possibility of improving accuracy in detecting small objects (flowers). The proposed method aims to replace manual flower counting, a process important for crop management.

Overall, I consider the article to be well-conceived, constructed, and presented. However, several points should be emphasized in the text:

Response: We are very appreciative of your professional review work on our article. We sincerely appreciate the positive comments. As you are concerned, all the amendments have been addressed. According to your suggestions, we have made extensive corrections to our previous draft, the detailed corrections are listed below.

Q1. Limited source dataset size: The dataset, consisting of 20 video clips and 1,421 images captured using a single device, may limit the applicability of the solution to that specific hardware (camera). It is important to stress the need for a more generalized solution by testing the model's performance on images captured with different exposure parameters (including using other smartphones) or acquiring training data from various devices. The high precision achieved might result from overfitting the model to a homogeneous dataset, which could be a case of cherry-picking. 

Response: Thank you for your valuable suggestions. Based on your suggestions, we have made modifications and additions in the discussion section, as follows:

Line 586-592: However, due to device limitations, we only use mobile phones to collect data, which may limit the applicability of this solution to specific hardware such as cameras. It is important to emphasize the need for more universal solutions by testing the performance of the model on different exposure parameters, including using other smartphones, capturing images, or obtaining training data from various devices. Otherwise, the high accuracy of model parameters trained on a single dataset may be due to overfitting of the model.

Q2. Lack of reference to image acquisition conditions: The article only specifies the date range during which the images were captured. It would be valuable to include a discussion of how factors like time of day, rain, sunlight, fog, humidity, etc., could affect the accuracy of the measurements. This element should be added to the discussion to address potential external environmental influences.

Response: We greatly appreciate your valuable suggestions. As per your recommendations, we have carefully revised the article and made the following changes:

Line 585-586: Meanwhile, factors such as time of day, rain, sunlight, fog, and humidity also have an impact on the accuracy of image collection.

Line 592-595: Therefore, improving photography conditions, enriching dataset sources such as collecting data in different external environments, and utilizing multiple devices including different models of mobile phones and cameras can help overcome these challenges.

Q3. Insufficient discussion of occlusion issues: The problem of occlusion or obstruction is only briefly mentioned. In practice, this is an important challenge that the model needs to handle. This should be further discussed, explaining how the method addresses situations where flowers are partially or completely hidden from the camera.

Response: Thanks for your good suggestions. Based on your suggestion, we have made the following modifications and text additions to discuss more about explaining how the method addresses situations where flowers are partially or completely hidden from the camera.

Line 572-580: Besides, instance segmentation techniques offer a refined approach to precisely delineate occluded flowers and discern their backgrounds, thereby enhancing the accuracy of identification. Unmanned Aerial Vehicle (UAV) and hyperspectral technology can also be used to detect rapeseed flowers. Nevertheless, the implementation of such methods necessitates a substantial investment in both human labor and material resources, including the annotation of large datasets by experts, the procurement of high-performance computational infrastructure, and the development of sophisticated algorithms capable of handling complex occlusions and background variations. 

Q4. Why StrongSORT was chosen: While the StrongSORT method for object tracking is described, it would be beneficial to explain why this particular method was chosen. Were any tests performed using other tracking methods? A comparison with alternative techniques would strengthen the justification for using StrongSORT.

Response: Thanks for your good questions. In the article, we tested other tracking methods like DeepSORT and compared them with StrongSORT as follows:

Line 435-452: A comparative analysis was performed against the current multi-target tracking algorithm DeepSORT, which also adopts the Deep Appearance Continuity paradigm. The results are documented in Table 6. The comparison reveals that in comparison with DeepSORT, StrongSORT showcased a notable increase in the MOTA index by 4.9%, HOTA by 2.8%, and IDF1 by 7.8%. This enhancement can be attributed to the refined trajectory and target matching process in StrongSORT, incorporating both appearance and motion characteristics, alongside advancements in the matching process and Kalman filtering.

While DeepSORT also leverages appearance and motion characteristics, its approach to appearance feature extraction is relatively straightforward, with motion features primarily utilized to eliminate implausible matches, leading to excessive reliance on appearance features. Moreover, the similar appearance between rapeseed flowers and flower buds can exacerbate identity-switching issues. Notably, all multi-target tracking algorithms in this study employ GhP2-YOLO as the detector, resulting in minimal differentiation between the two algorithms in terms of MOTP metrics. DeepSORT and StrongSORT exhibit lower MOTP metrics due to the target box output of DeepSORT post the Kalman filter update stage, impacting the box position based on Kalman filtering adjustments.

Table 6. Metrics of different multi-object tracking algorithms for tracking test videos.

MOT algorithm

MOTA/%

HOTA/%

IDF1/%

DeepSORT

64.5

54.3

66.5

StrongSORT

69.4

57.1

74.3

Meanwhile, we explained why the StrongSORT method was ultimately chosen for target tracking in rapeseed flower detection as follows:

Line 541-553: StrongSORT is chosen as the target tracking method because it excels in multi-target tracking by combining advanced deep learning feature extraction with an efficient online tracking-detection framework. It effectively handles complex scenarios, such as target occlusion, fast movements, and interactions between multiple targets, ensuring stability and accuracy. For applications requiring both real-time performance and high precision, StrongSORT is the ideal choice, seamlessly integrating deep features with advanced tracking algorithms for smooth and reliable multi-target tracking. The StrongSORT was chosen for target tracking in rapeseed flower detection due to its excellence in multi-target tracking. It combines advanced deep learning feature extraction with an efficient online tracking-detection framework, making it ideal for handling fast movements and interactions between multiple flower targets in the video. StrongSORT ensures stability and accuracy, making it the preferred choice for applications demanding both real-time performance and high precision.

Q5. Field application of the GUI: Although the article touches on the topic of the application's interface, it does not sufficiently address its utility in field conditions. How well does the system function in agricultural environments? If everything is processed in post-processing, this should be clearly stated. Additionally, the equipment used (GPU RTX 3060) does not seem to be practical for widespread use in (agriculture fieldwork). Further clarification of the practical utility of the solution in real-world agricultural settings is needed.

Response: We greatly appreciate your good questions and professional advice. Based on your proposal, we have made the following explanations and modifications:

Line 512-519: Our endeavors primarily revolve around the seminal exploration of harnessing deep learning paradigms for the recognition and enumeration of rapeseed blossoms, coupled with the development of an intricate visualization system geared towards real-time monitoring. Our research endeavors are meticulously crafted with an eye toward future applications, specifically targeting embedded devices like the Raspberry Pi. As we progress, our focus will intensify on refining and optimizing these solutions, ensuring their seamless integration and efficacy in practical scenarios.

In addition, our experimental device (GPU RTX 3060) is not the most powerful computing device. The GhP2-YOLOm model with the largest architecture we trained has only 14.69M parameters and can also be applied on ordinary CPU devices. In the future, we will compress the model on device terminals such as Raspberry Pi and apply it to real-life agricultural environments.

Positive Aspects of the Article:

Despite the mentioned concerns, I would like to highlight several positive aspects of the article:

  • The article presents a promising automated method for flower counting, which has significant potential to improve the efficiency of crop management.

  • According to the results obtained and presented by the authors, the accuracy of the method is very high. However, attention should be given to the potential issue of model overfitting.

  • The article demonstrates a good reduction in the base CNN model's complexity through the application of the Ghost module, making the approach more computationally efficient.

  • There is potential for extending the method to other types of flower clusters and additional agricultural applications, suggesting a broad applicability for the proposed system.

Therefore, after introducing a few revisions in the text, I consider the reviewed article worthy of publication.

Response:

We greatly appreciate your excellent suggestions and positive comments, especially your efforts in providing detailed revision tips, which are very useful in helping us correct errors and improve the quality of our manuscript. Based on your suggestions, we have carefully revised the manuscript to meet the publication standards.

Reviewer 2 Report

Comments and Suggestions for Authors

Comments to Authors

Can you explain in detail during the flowering phase accurate quantification of flora and their anatomical structures is important for botanical breeders and agricultural cultivators, particularly?

Are there any other methodologies for the quantification of specific plant components, such as flowers, among different genotypes, I suggest mentioning them also.

What is the difference between the features of the GhP2-YOLO neural network, and traditional YOLO v8 object detection model.

Can you explain in detail how the StrongSORT algorithm contributes to the tracking of rapeseed flower buds and blossom patterns within video datasets.

What is the significance of the R-squared coefficients (0.9719, 0.986, and 0.9753) obtained in the comparative analysis of manual and automated counts of rapeseed flowers and buds, if they are so important there are several other related parameters like SSE, MSE, RMSE provide and investigate how they are related with each other.

How this study can be used for the use of deep convolutional neural networks and multi-target tracking protocols development when concerning botanical research and agricultural administration?

Minor

After Fig 7, there will be Fig 8 not 9 correct it.

Improve the font size of legends of Fig 7 and 9.

And change the text accordingly.

Author Response

From the reviewer #2

Comments and Suggestions for Authors:

#1 Can you explain in detail during the flowering phase accurate quantification of flora and their anatomical structures is important for botanical breeders and agricultural cultivators, particularly?

Response: Thank you for your good question. The explanation for your question is as follows:

Line 508-511: During the flowering phase, precise quantification of flora and their anatomical structures is paramount for botanical breeders and agricultural cultivators, as it informs selective breeding programs, enhances crop resilience, and optimizes yield potential by understanding the intricacies of plant reproduction and growth.

#2 Are there any other methodologies for the quantification of specific plant components, such as flowers, among different genotypes, I suggest mentioning them also.

Response: Thank you for your insightful and thoughtful questions. We have provided several simple examples to illustrate the research on the classification of different genotypes of crops by different plant components.

Line 75-77: Estrada et al. used high-density images based on deep learning to detect and count the number of flowers in peach groves, which can serve as an early prediction indicator for crop yield[21].

Line 77-79: Bi et al. proposed the ZGWO-RF model for non-destructive classification of 11 maize seeds based on multi-source data, which preliminarily explored a method for quantifying specific plant components in different genotypes[22].

Line 91-92: Feng et al. presented a lightweight YOLOv7-based citrus detection and dynamic counting approach, tailored for modern planting scenarios in citrus image and video analysis[24].

#3 What is the difference between the features of the GhP2-YOLO neural network, and traditional YOLO v8 object detection model.

Response: Thanks for your good questions. I am sorry for not highlighting the parts you mentioned in our previous article. The features between GhP2-YOLO and YOLO v8 are that GhP2-YOLO detection speed is faster, the model is lighter, and the accuracy is higher than YOLO v8. We have made modifications and illustrations according to your suggestions. The supplement of this paper is as follow:

Line 167-171: The introduction of the P2 structure in the network head aims to improve the detection of small objects, while the incorporation of Ghost modules in place of a few Conv modules in both the backbone and head strives for model lightweight and reduced parameter complexity. Simultaneously, some C2f modules have been replaced by C3Ghost modules. This enhanced model, known as GhP2-YOLO.

Line 252-255: In summary, GhostConv introduces a channel separation mechanism in Conv, which can divide a channel into several sub channels to obtain more information for free without consuming additional computing resources, ensuring that the number of parameters remains unchanged or even reduced.

#4 Can you explain in detail how the StrongSORT algorithm contributes to the tracking of rapeseed flower buds and blossom patterns within video datasets.

Response: Thank you for your excellent question. I would explain in detail in the following text how the StrongSORT algorithm helps track rapeseed flower buds and flowers in video datasets, and supplement it in the manuscript:

Line 541-553: StrongSORT is chosen as the target tracking method because it excels in multi-target tracking by combining advanced deep learning feature extraction with an efficient online tracking-detection framework. It effectively handles complex scenarios, such as target occlusion, fast movements, and interactions between multiple targets, ensuring stability and accuracy. For applications requiring both real-time performance and high precision, StrongSORT is the ideal choice, seamlessly integrating deep features with advanced tracking algorithms for smooth and reliable multi-target tracking. The StrongSORT was chosen for target tracking in rapeseed flower detection due to its excellence in multi-target tracking. It combines advanced deep learning feature extraction with an efficient online tracking-detection framework, making it ideal for handling fast movements and interactions between multiple flower targets in the video. StrongSORT ensures stability and accuracy, making it the preferred choice for applications demanding both real-time performance and high precision.

#5 What is the significance of the R-squared coefficients (0.9719, 0.986, and 0.9753) obtained in the comparative analysis of manual and automated counts of rapeseed flowers and buds, if they are so important there are several other related parameters like SSE, MSE, RMSE provide and investigate how they are related with each other.

Response: Thanks for the valuable suggestions. The significance of the R-squared coefficient obtained in the comparative analysis of manual and automatic counting of rapeseed flowers and buds is to compare the differences between the two counting methods, with the aim of proving the feasibility of our proposed automatic counting method. Then, based on your suggestion, we have added several other relevant parameters to our manuscript, such as SSE, MSE, and RMSE. According to your suggestion, we have made the following modifications and supplementations as follows:

Line 460-463: Table 7 presents the comprehensive statistical data comparing the actual measurement and machine counting of rapeseed, encompassing the Sum of Squared Errors (SSE), Mean Squared Error (MSE), Root Mean Squared Error (RMSE), and the coefficient of determination (R²).

Line 468-472: Drawing upon the data presented in Table 7 and Figure 8, a conclusion can be drawn that there exists a strong correlation between the manually and machine-measured counts of rapeseed flower buds, flowers, and the total number. This observation underscores the feasibility of employing deep learning models for the purpose of accurately counting rapeseed flower buds and flowers.

Table 7. Several statistical data on rapeseed flower buds, flowers, and total

SSE

MSE

RMSE

R2

Buds

104

5.2

2.280351

0.986

Flowers

275

13.75

3.708099

0.9719

Total

349

17.45

4.17732

0.9753

#6 How this study can be used for the use of deep convolutional neural networks and multi-target tracking protocols development when concerning botanical research and agricultural administration?

Response: Thanks for your good question. According to your question, we have made the following corrections and explanations:

Line 44-47: These special traits can serve as indicators of rapeseed yield, which ultimately plays a pivotal role in shaping breeding strategies and may also offer farmers invaluable insights for devising hedging tactics that ensure crop stability and profitability.

Line 596-600: This research significantly facilitates mechanized and intelligent field management, thereby reducing human labor costs and significantly enhancing farmers' planting efficiency and productivity. By embracing advanced technological solutions like deep learning, it paves the way for more streamlined agricultural operations and ultimately boosts the profitability of farming endeavors.

Minor

#7 After Fig 7, there will be Fig 8 not 9 correct it.

Improve the font size of legends of Fig 7 and 9.

 And change the text accordingly.

Response: I am grateful for your meticulous review and invaluable insights, which have guided us in rectifying the erroneous numbering and enhancing the image's legibility through an enlarged font. Your contributions have undoubtedly enriched the overall presentation.
